# Environmental Effect on Fatigue Crack Initiation under Equi-Biaxial Loading of an Austenitic Stainless Steel

**Cédric Gourdin [1,2,\*], Grégory Perez [1], Hager Dhahri [1,2], Laurent De Baglion [3]** 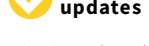 **and Jean-Christophe Le Roux [4]**

[1] CEA, Services d'Etudes Mécaniques et Thermiques (SEMT), Université Paris-Saclay, F-91191 Gif-sur-Yvette, France; gregory.perez@cea.fr (G.P.); hager.dhahri@cea.fr (H.D.)
[2] IMSIA, UMR 9219, CNRS, CEA, EDF, Université Paris-Saclay, 91762 Palaiseau CEDEX, France
[3] FRAMATOME SAS, Tour AREVA, F-92084 Paris La Défense, France; laurent.debaglion@framatome.com
[4] EDF, R&D, Site des Renardières, F-77818 Moret sur Loing CEDEX, France; jean-christophe.le-roux@edf.fr
[\*] Correspondence: cedric.gourdin@cea.fr; Tel.: +33-1-69-08-45-86

**Abstract:** The lifetime extension of nuclear power stations is considered an energy challenge worldwide. That is why the risk analysis and the study of various effects of different factors that could potentially prevent safe long-term operation are necessary. These structures, often of great dimensions, are subjected during their life to complex loading combining varying multiaxial mechanical loads with non-zero mean values associated with temperature fluctuations under a PWR (pressure water reactor) environment. Based on more recent fatigue data (including tests at 300 °C in air and a PWR environment, etc.), some international codes (RCC-M, ASME, and others) have proposed and suggested a modification of the austenitic stainless steels fatigue curve combined with a calculation of an environmental penalty factor, namely Fen, which has to be multiplied by the usual fatigue usage factor. The determination of the field of validation of the application of this penalty factor requires obtaining experimental data. The aim of this paper is to present a new device, "FABIME2e" developed in the LISN (Laboratory of Integrity of Structures and Normalization) in collaboration with EDF (Electricity of France) and Framatome. These new tests allow the effect of a PWR environment on a disk specimen to be quantified. This new device combines structural effects such as equibiaxiality and mean strain and the environmental penalty effect with the use of a PWR environment during fatigue tests.

**Keywords:** fatigue; multi-axial fatigue; environmental fatigue; experimental; fatigue life criteria; Fen; austenitic stainless steel

## 1. Introduction

The question of assessing the margins and safety factors in fatigue analyses, which are widely used today (ASME BPV III, RCC-M, JSME, EN-13445-3, etc. [1–5]), is a very challenging one.

The fatigue rules used today in the nuclear industry were initially built and integrated into the ASME code in the 1960s. Establishing fatigue rules is a challenge in itself since fatigue degradation depends on the wear of components that undergo repeated cycling. Fatigue tests can therefore be very long and costly, if led on full-size components. As a result, testing is, in practice, conducted on small laboratory specimens, which then raises the question of how to extrapolate results to a full-size component. Another difficulty is that the rules need to remain easy to apply in order to be used in industrial engineering calculations. Since 2007, the USA, with NUREG/CR-6909 [1], has included the evaluation of environmental effects in their official regulation. Indeed, on the curves presented in the NUREG report [1], the PWR water environment effect on the fatigue lifetime of material used in the manufacture of reactor components is illustrated.

The 304L and the 316L stainless steels are used for the manufacture of pressurized water reactors (PWR). Many components of this type of reactor are subjected to a multiaxial

thermo-mechanical cycling [6]. Therefore, the multiaxial fatigue assisted by the environment is considered one of the possible degradation mechanisms affecting the life of the PWR components.

Based on more recent fatigue data (including tests at 300 °C in air and PWR environment, etc.), some international codes (RCC-M, ASME, and others) have proposed and suggested a modification of the austenitic stainless steel fatigue curve combined with a calculation of an environmental penalty factor, namely Fen, which has to be multiplied by the usual fatigue usage factor.

Unfortunately, testing on structures representative of real plant components is expensive but should be increased to help contribute to the general understanding of the various aggravating effects [7–14]. In order to obtain fatigue strength data under structural loading, biaxial test devices with and without a PWR environment were developed at LISN [15–19].

The basic idea of the disc bending fatigue test was presented by Ives et al. [11] and Shewchuk et al. [12] about 50 years ago. In this test technique, a disc specimen is subjected to bending load by applying air pressure on the specimen surface. By altering the constraining condition at the edge of the specimen, a crack can be initiated at the specimen center even when a uniform thickness specimen is used, but the specimen diameter is more than 250 mm [13].

Two kinds of fatigue devices have been developed. Within the same specimen geometry, structural loads can be applied by varying only the PWR environment. The first device (FABIME2) is devoted to studying the effect of biaxiality and mean strain/stress on the fatigue life. A second and new device, named FABIME2e and based on the first device, is for the study of the impact of the environmental effects. With these new experimental results, the study of the interaction between PWR environment and multiaxial loading is achievable.

## 2. The Equibiaxial Fatigue Device with Oil Environment

Reference uniaxial fatigue tests were performed with load ratios equal to −1. This means that the uniaxial specimen was subjected to the same level of loading in both tension and compression.

The objective here was to carry out fatigue tests on a non-standard specimen, but representing an intermediate step between uniaxial specimens and complex structures on which the fatigue criteria were applied, with the same type of mechanical loading. This means that the device must be able to apply a load with a load ratio that can vary from 0 to −1, i.e., a purely repeated loading with completely alternating loading. The load ratio is defined as the ratio of the minimum (in displacement) load to the maximum load at the most loaded surface point of the specimen's useful area.

For this purpose, the principle of spherical bending was applied. The advantage of the spherical bending test specimen is that it allows the application of an equibiaxial strain in the plane that is, for example, representative of a deformation resulting from thermal loading. The application of spherical bending was achieved by subjecting a circumferential specimen to a pressure differential between its two faces (Figure 1).

In order to concentrate the equibiaxial deformations in the central area of the specimen, an optimization of the geometry was carried out, and more particularly the definition of the thickness of the specimen.

The spherical bending specimen was positioned in a test cell, which is the active part of the FABIME2 experimental device (Figure 2) (Top Industry, Vaux Le Penil, France).

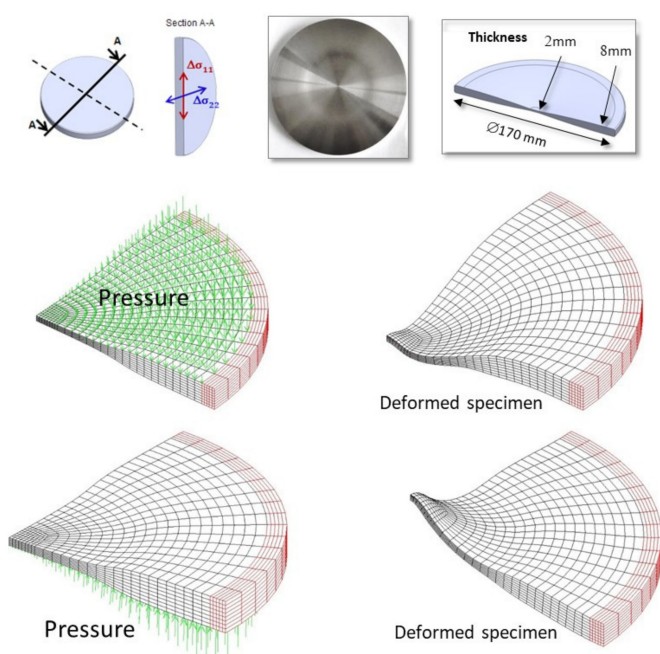

**Figure 1.** Principle of the equibiaxial fatigue test.

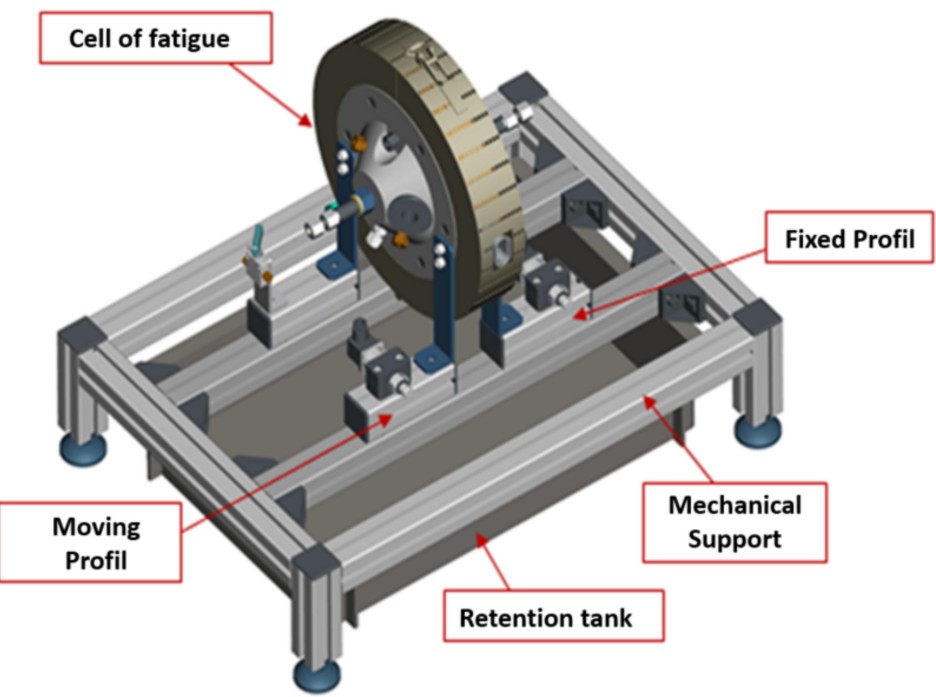

**Figure 2.** View of the spherical bending device fatigue cell.

This cell was composed of two half-shells enclosing the fatigue specimen. The test cell was connected to an oil pressure generator (Top Industry, Vaux Le Penil, France) with a maximum capacity of 100 bars.

The control system was developed at LISN with the LABVIEW software (v.2016, National Instrument, Austin, TX, USA). This dedicated CEA (French Alternative Energies and Atomic Energy Commission) in-house software allows fatigue tests to be carried out in two different control modes:

- Either in imposed displacement: by measuring the deflection in the center of the specimen via LVDTs (linear variable differential transformers) positioned in the center of the specimen; or
- In imposed pressure: by a measurement of the pressure level in the chamber of the fatigue cell.

Oil temperature measurements (type K thermocouple, Hydac, Sulzbach/Saar, Germany) in the fatigue cell chamber complete the instrumentation of the device.

The maximum deflection that can be applied is ±15 mm. Depending on the level of displacement load, the ranges of the LVDTs used to measure deflection during testing are either ±5 mm, ± 10 mm, or ±15 mm.

During the development phase of the device and also after each fatigue test, surface observations were made to ensure that there was no influence of the presence of the LVDT (contact area) on the crack initiation sites.

The use of a translucent oil allowed the surface of the equibiaxial fatigue specimens to be observed during the tests. Photographs were taken through portholes (borosilicate glass) oriented at 45° with a diameter of 20 mm.

European Security directives (Machines 2006/42/CE, Pression 97/23/CE) were respected.

## 3. The Experimental Results from FABIME2 and Interpretation

Biaxial fatigue tests were carried out on two austenitic stainless steels: "316L THY" and "304L CLI." The first material was provided by Thyssen Krupp Materials France as a 15 mm thick rolled sheet. The second material supplied by EDF was characterized by a 30 mm thick rolled sheet.

The application of design rules to prevent fatigue damage, as defined in the nuclear design rules (RCC-M, RSE-M and RCC-MRx), was based on the use of equivalent quantities characterizing the mechanical state. These equivalent quantities of stresses and strains were described using either the von Mises criterion (RCC-MRx) or the Tresca criterion (RCC-M and RSE-M).

The description of the mechanical state with these equivalent quantities makes it possible to apply the fatigue data obtained on standard uniaxial specimens to structures with more complex geometries.

Thus, the first equivalent strain used was the von Mises equivalent strain, which is defined by the following equation:

$$\Delta\varepsilon_{eq}^{VM} = \sqrt{\frac{1}{1+v'}(\Delta\varepsilon_d : \Delta\varepsilon_d)} = \frac{2}{3}\frac{(1+v')}{(1-v')}\Delta\varepsilon_1 \tag{1}$$

with the deviatoric strain component:

$$\Delta\varepsilon_d = \Delta\varepsilon - \frac{1}{3}tr(\Delta\varepsilon) \tag{2}$$

where $\varepsilon_1$ is the principal strain and $v'$ the "real" Poisson ratio (elastic $v = 0.3$ and plastic part $v = 0.5$).

The second equivalent strain was the Tresca equivalent strain, which is defined by the following equation:

$$\Delta\varepsilon_{eq}^{T} = \frac{1}{1+v'}Max|\varepsilon_i - \varepsilon_j| \tag{3}$$

The interpretation of the equibiaxial fatigue tests carried out in this study could not be made directly, and required an analysis based on two hypotheses.

The first one was of an experimental nature: It allowed us to link the displacement imposed in the center of the specimen to the radial strain. This experimental calibration step is described in the following reference [19]. At the end of this first step, an experimental

calibration curve, carried out on specimens of the same material as the one to be studied, was thus defined.

The second one was purely numerical. It allowed the radial strain to be related to the equivalent magnitude of the strains (von Mises or Tresca equivalent) by taking into account the plastic part of the material's behavior. Numerical simulations with elasto-plastic behavior models of fatigue tests allowed us to determine this relationship between radial strain and equivalent strain.

A test campaign was carried out on two austenitic stainless steels (316L and 304CLI), under symmetrical displacement loading conditions (i.e., with a load ratio of R = −1). The tests were interpreted according to the proposed interpretation methodology, and the results obtained were compared with the results of standard uniaxial reference fatigue tests (Figure 3).

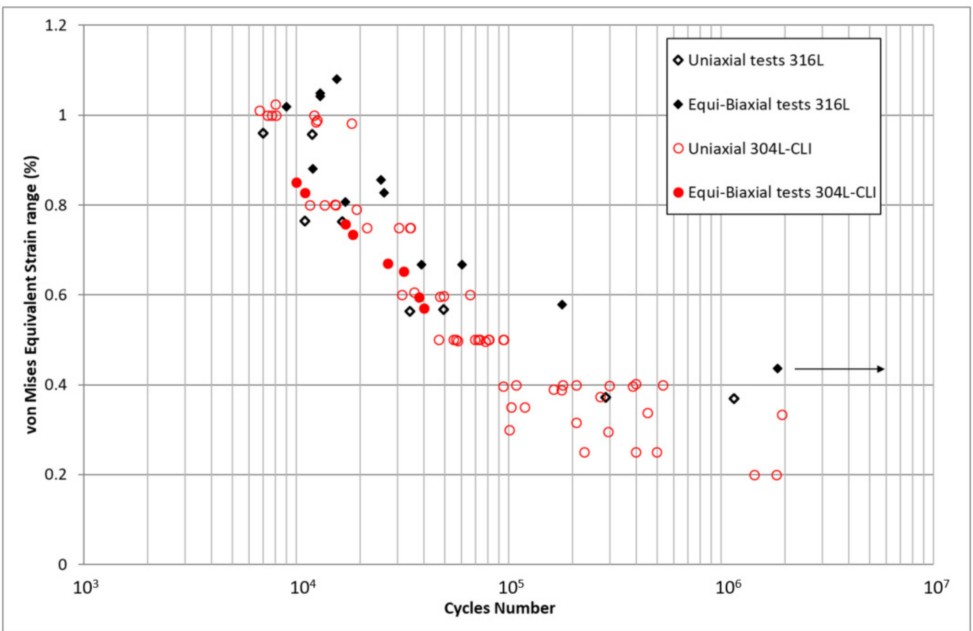

**Figure 3.** Fatigue data obtained on the two austenitic stainless steels (316L and 304CLI).

The change in fatigue life with the von Mises equivalent strains defined previously is shown in Figure 3. The equivalent strain range was defined by the use of the relation between the radial strain and the equivalent strain. This relation was determined by elasto-plastic calculation of the fatigue test. It appears that there was no significant impact of equibiaxial fatigue for the two types of materials, considering von Mises equivalent strains.

## 4. The Equibiaxial Fatigue Device with PWR Conditions: FABIME2e

### 4.1. Presentation of the Equibiaxial Fatigue Device with PWR Conditions: FABIME2e

The study of the aggravating effects of the PWR environment requires fatigue tests with and without this aggravating effect. The other characteristics of the fatigue tests (in terms of geometry, loading conditions, etc.) must be as identical as possible.

The main characteristics of the PWR environment are:

- A higher operating temperature, of the order of 340 °C;
- A higher pressure to ensure that the environment remains monophasic, of the order of 150 bars;
- A control of the chemistry of the environment and in particular the level of dissolved hydrogen;
- The specimen is in permanent contact with the PWR environment.

The adaptations and modifications of the fatigue device with an oil environment in order to carry out fatigue tests under PWR conditions mainly concern three technological points to be solved. The first concerns the possibility of applying mechanical loading via the PWR environment. The second concerns the precise management and control of the temperature of the environment. The third addresses the problem of measuring the various physical quantities necessary for the proper conduct and analysis of the tests.

A double cylinder system was proposed to separate PWR and hydraulic fluids to apply mechanical loading to the specimen (Figure 4). A double acting cylinder is moved by the hydraulic unit. Its movement is mechanically transmitted (by the water incompressibility) to a primary cylinder to modify the volume of the PWR environment contained in each half-shell. Similar to FABIME2, this system applies a differential pressure of up to 100 bars to the specimen. The difference here is that the pressure variation (i.e., up to 100 bars) around the specimen can be applied within a pressure range of 150 to 350 bars, corresponding respectively to the biphasic threshold of the PWR environment and the maximum pressure allowed by FABIME2e.

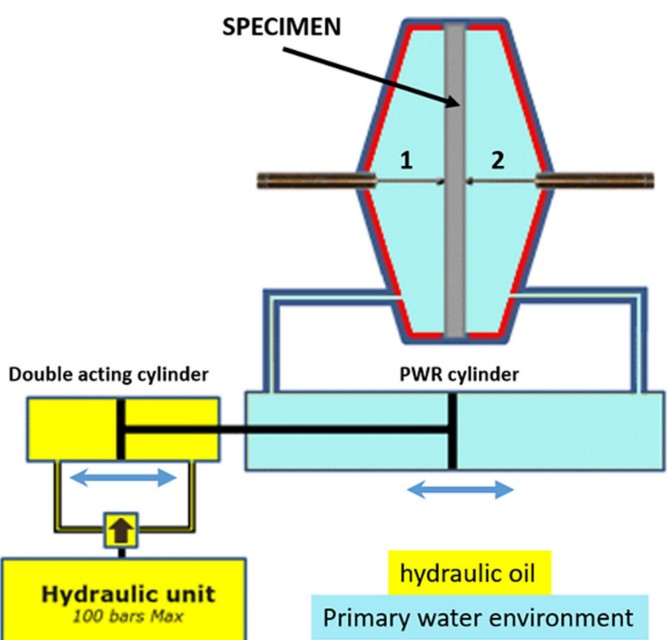

**Figure 4.** Double cylinder system for the separation of the PWR and hydraulic fluid.

The principle of detection of fatigue crack initiation is based on the method of the measurement of compliance. For example, during fatigue tests where the displacement is controlled and constant throughout the test, the detection of crack initiation is characterized by a change in the pressure level to be reached for said imposed displacement. However, the temperature has a strong impact on this global pressure level in the test chamber—even a small variation of about 1 °C causes changes in the pressure level up to a few bars. Precise temperature control must be guaranteed during the test and is a fundamental criterion for the smooth running of the tests, and above all to ensure good crack initiation detection capability.

The device is equipped with various sensors that allow the mechanical loading imposed via pressure and displacement sensors in the center of the specimen to be controlled. The characteristics of the environment must also be measured during the tests, including the temperature of the environment and the level of dissolved hydrogen.

Finally, it should be noted that ensuring good sealing of a circuit containing an environment at high temperature (340 °C up to 400 °C) and high pressure (up to 350 bars) is a technological challenge in itself.

### 4.2. Description of the Equibiaxial Fatigue Device: FABIME2e

The French company TOP INDUSTRY, specializing in equipment under high pressure and with aggressive environments, enabled us to design and build our experimental device: FABIME2e. A maximum pressure of 350 bars and a maximum temperature of 340 °C are the experimental capacities of our fatigue device.

A complete view of the experimental device is shown in the following figure (Figure 5). In it we find the fatigue test cell in the center of the test device. This cell integrates a sealing compatible with the PWR environment. This cell is connected to a circuit containing the PWR environment. The link between the circuit for applying mechanical forces and the circuit of the PWR environment is ensured by the intermediate cylinder. On the front panel of the device, there is a board allowing the level of dissolved hydrogen to be controlled and adjusted.

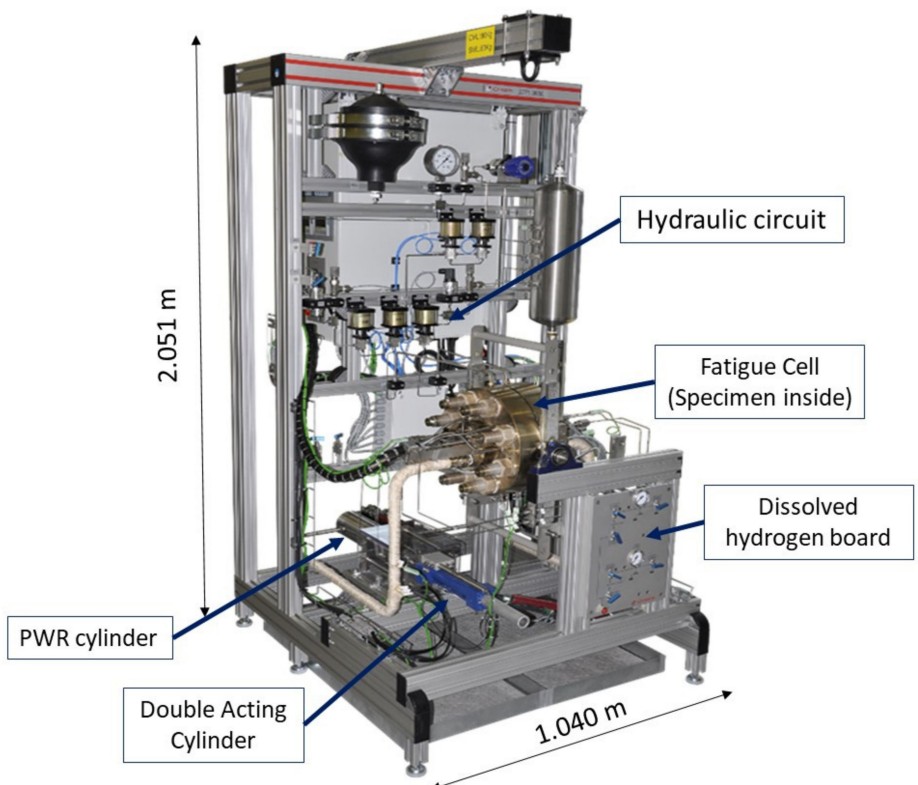

**Figure 5.** View of the new fatigue bending device FABIME2e.

The high-pressure level as well as the higher temperatures result in an increase in the dimensions of the test cell (Figure 6), even though the dimensions of the specimen remain the same for both test devices (FABIME2 and FABIME2e).

On each half cell there are several sensors, such as:

- Type K thermocouples allowing the temperature of the environment as well as the temperature gradient within it to be measured;
- Pressure sensors with a capacity of 400 bars;
- LVDT-type specimen center displacement sensors from RDPE Industry (these displacement sensors support PWR environment conditions and their measuring range is ±5 mm);
- Two Pd–Ag hydrogen sensors from Framatome: one for measuring and one for adjusting the dissolved hydrogen level if necessary.

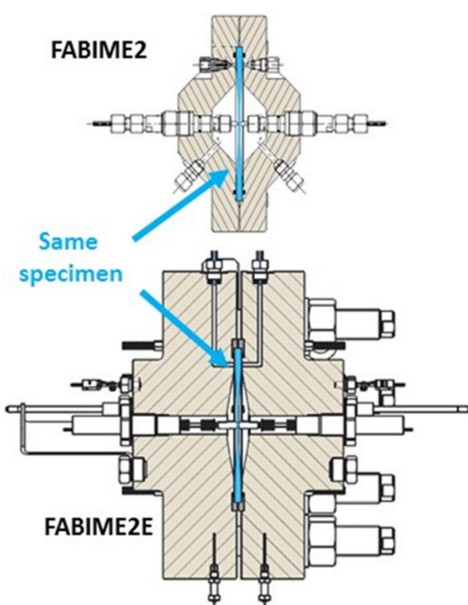

**Figure 6.** Comparison between FABIME2 and FABIME2e fatigue cells (same scale and specimen geometry, $\varnothing$ = 170 mm).

The sealing of the circuit containing the PWR environment was ensured by the use of metallic EYTOR O-rings. However, the use of this type of seal requires the use of a clamping system via the use of hydraulic tension cylinders. A procedure for the installation of the seals was defined and ensured a tightness by means of 8 heavy section studs.

The temperature setting as well as the maintenance of this one around 340 °C (for a maximum temperature of 400 °C) during the fatigue tests was carried out by means of 8 heating cartridges with a power of 250 W each. In order to avoid sudden temperature variations, the temperature rise speed was limited to 1 °C/min. The heating system was regulated by two Eurotherm controllers (Nanodac model). Only the test cell and the supply pipes directly attached to the cell were kept at the right temperature. The rest of the circuit was at room temperature, especially the intermediate cylinder.

The development of the control software in the CEA laboratory allowed great flexibility: cycling shape, holds, control mode modifications, and mean pressure or strain. The low-level tasks such as security management, hydraulic control, and data reading required determinism and speed of processing. That is the reason why they were devolved to real-time autonomous software running on a CompactRIO device (National Instruments). The test management, acquisition, and data analysis were performed by software running on a conventional PC. This second software controlled each test sequence: from the filling of the PWR fluid to the crack initiation estimation by sending orders to the CRIO software and the Eurotherm controllers.

## 5. First Equibiaxial Fatigue Tests under Distilled Water Conditions

Equibiaxial environmental fatigue test devices implement two characteristics. The first is the use of the incompressibility of the environment and the second is a physical separation of the environments. The transmission of mechanical energy from one environment to the other is achieved by means of an intermediate cylinder. The first phase of validation of these test devices consisted of verifying this concept.

Next, a calibration of the sensors and the detection capability of crack initiation are detailed.

### 5.1. Deflection Calibration

Before each fatigue test campaign, it was necessary to check and calibrate the various sensors enclosed in the fatigue device. However, the calibration of the displacement

sensors was more delicate to carry out, especially in high temperature conditions (300 °C). A calibration specimen was specially designed for this purpose. This had a sliding element that allowed calibrated displacements to be imposed on the LVDTs (Figure 7).

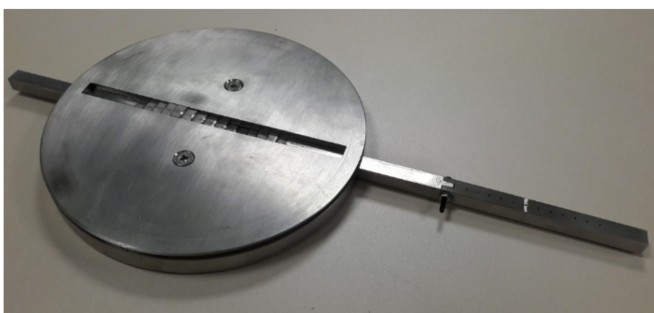

**Figure 7.** Dedicated calibration specimen used to calibrate the LVDT at ambient temperature and at 300 °C.

The capability of the device to impose displacements at fixed rates was also verified during the validation phase of the fatigue test device (Figure 8).

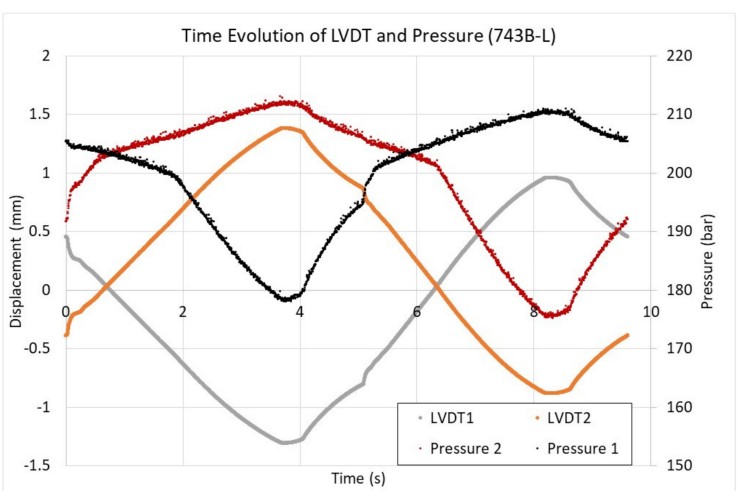

**Figure 8.** Time evolution of the displacement (LVDT) and the applied pressure for a spherical fatigue test.

### 5.2. Discussion of the Crack Initiation Criteria

The particularity of the FABIME2 fatigue specimen is to allow fatigue tests to be carried out on an intermediate scale between reference tests on uniaxial standard specimens and tests on real or more complex structures. The criterion of fatigue life used in engineering and design methods is classically called $N_{25}$. This criterion corresponds to a reduction of the applied uniaxial stress by 25% during fatigue tests with imposed deformation. This $N_{25}$ criterion can be related to a 3 mm depth crack in an 8 to 9 mm diameter cylindrical specimen. The comparison of fatigue test results between equibiaxial tests on FABIME2 and reference fatigue tests required the definition of an $N_{25}$ criterion of fatigue life applicable to FABIME2 specimens. The mechanical state between the uniaxial and equibiaxial tests was not identical, the applied stress was oriented along the stress axis in the case of uniaxial specimens, whereas it was along two directions in the case of the equibiaxial tests (see Figure 9).

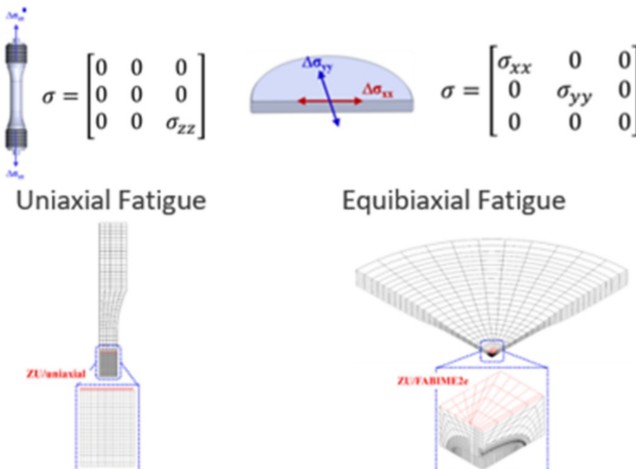

**Figure 9.** Difference stress tensor on uniaxial and equibiaxial specimens.

Numerical simulations of the FABIME2 tests were carried out with the FE Software Cast3M. Different semi-elliptical crack sizes were studied, starting from a solid specimen up to a crack size of 10 mm (dimension of the useful area) (see Figure 10). For each crack size, the simulations were performed, taking into account the nonlinearity of the material behavior using an elasto-plastic behavior model stabilized with nonlinear isotropic and kinematic strain-hardening.

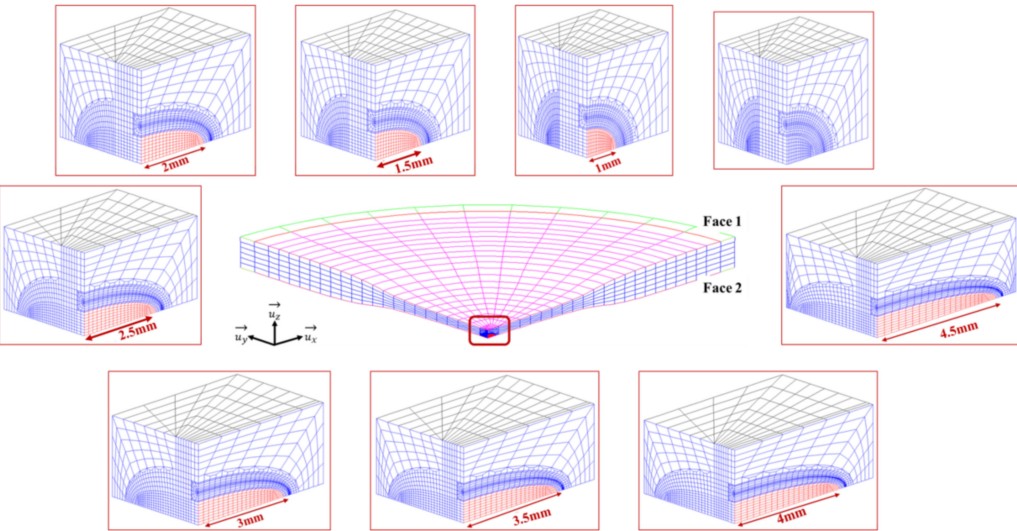

**Figure 10.** Mesh of the numerical model used for different crack sizes.

From the results obtained, the stress normal to the plane containing the crack could be extracted and the decrease of this stress with the size of the crack could be determined. The results for two loading levels are shown in the following figure (see Figure 11). Thus, for a surface crack size of 5 mm, the applied stress decreased by 25%.

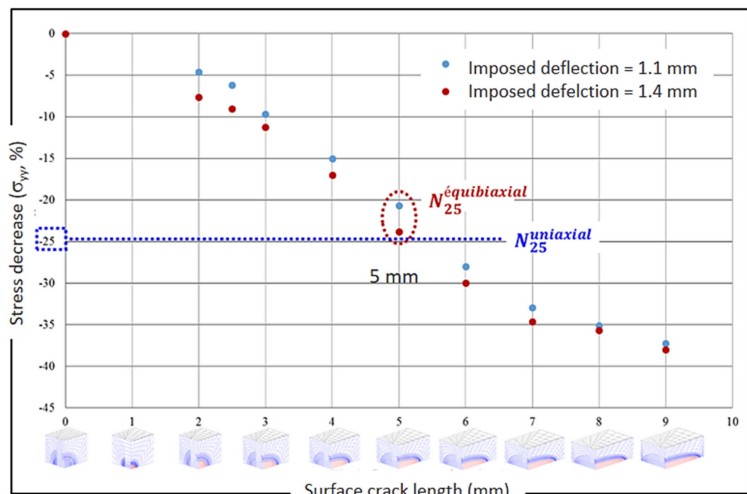

**Figure 11.** Evolution of the normal stress in the plane of the applied crack as a function of the size of the crack.

Experimentally, this decrease in applied stress (normal to the plane of the crack) led to a decrease in the pressure needed to obtain the same level of loading. The principle of the detection of fatigue crack initiation was achieved by studying the evolution of the pressure applied to the fluid via the intermediate cylinder to obtain the imposed deflection. When a "significant" change in this pressure was observed (Figure 12), the test was stopped, and the test specimen was take off and observed. The crack initiation criterion was defined as having a crack length of 5 mm at the surface on the observed side (Figure 13), which corresponded to a (downward) variation in the pressure of the order of 1 to 2 bars.

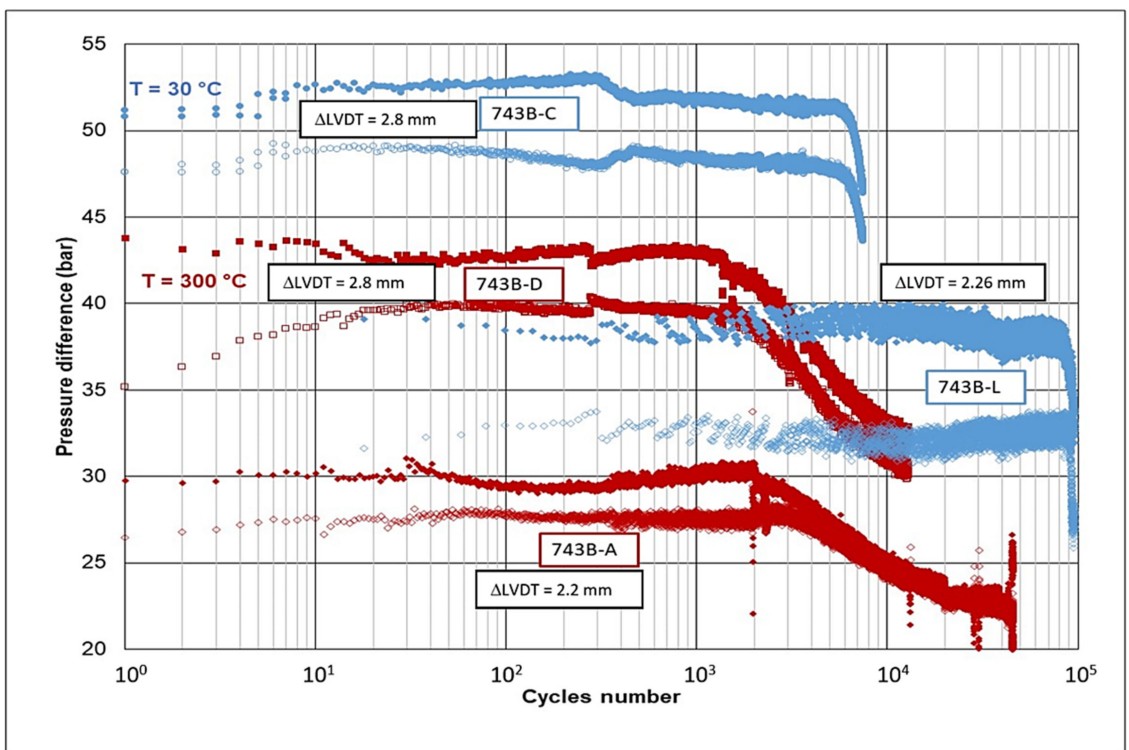

**Figure 12.** Time evolution of pressure differences in each chamber of the test cell for four tests (tests at a temperature of 30 °C = blue curves, tests at a temperature of 300 °C = red curves).

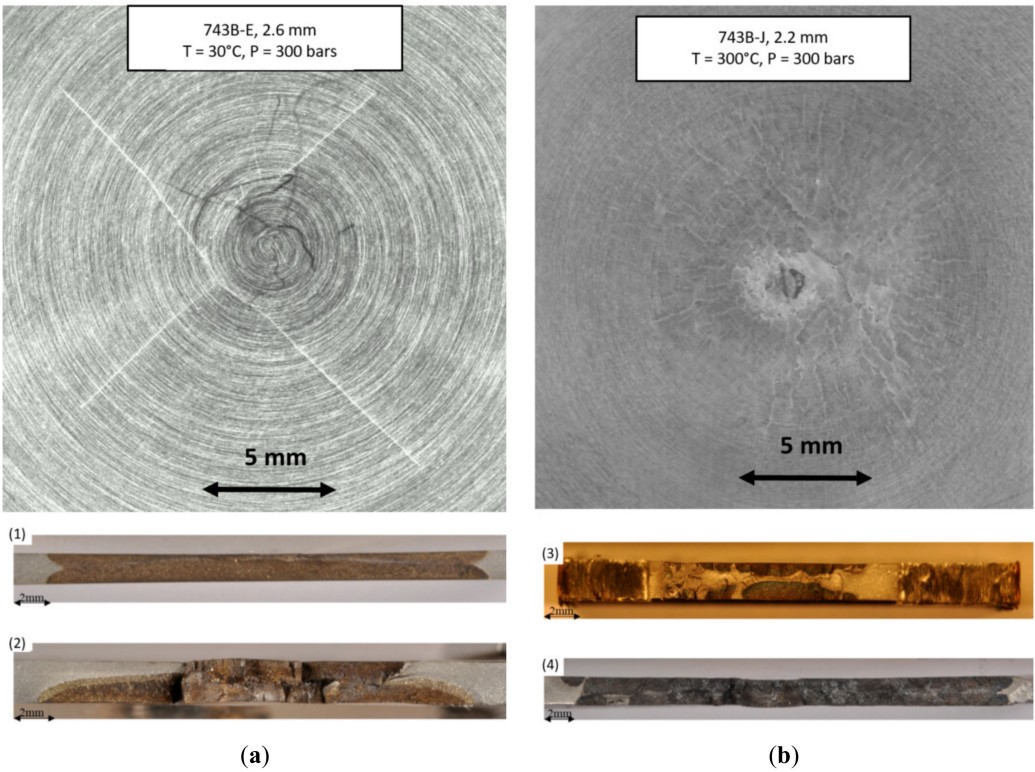

**Figure 13.** View of the fatigue crack initiation on the spherical bending specimen (**a**) without the water effect (at 30 °C) and (**b**) with the water effect (at 300 °C).

### 5.3. Experimental Results of Distilled Water Conditions

Equibiaxial fatigue tests on distilled water conditions were performed by imposing a constant deflection (three levels of deflection were imposed). Different levels of initial pressure (50 and 250 bars) and different levels of temperature (30 and 300 °C) were tested.

In Figure 14, the evolution of the pressure differential for four tests is shown. The tests performed at room temperature (30 °C, 743B-C for 2.8 mm deflection, and 743B-L for 2.26 mm deflection) were compared to the tests performed at 300 °C (743B-D for 2.8 mm deflection and 743B-A for 2.2 mm deflection). The obtained results show that, for the same level of imposed deflection, the pressure level was higher at room temperature, which can be explained by the difference in the elastounder distilled water conditions) was observed. For example, for tests with an imposed deflection variation of 2.8 mm, the number of cycles to initiate a 5 mm crack on the surface with no environmental effect was 8500 cycles, whereas it was between 2500 and 6000 cycles with distilled water conditions. The same type of result was also found for tests with a variation in deflection of 2.4 mm (from 18,500 cycles to 4000 cycles under distilled water conditions).

European programs (INCEFA+) or tripartite projects (with EDF and Framatome) provide a large number of experimental data. The aggravating factors are numerous but it is possible to classify them according to their level of importance. Thus, in the first place, test campaigns were carried out in order to study and clarify the effect of the strain rate on the lifetime. Identifying these limits and proposing a rule to take this effect into account, especially on austenitic stainless steel types 304L and 316L, is a short-term challenge. At a lower level, but not negligible, one can add the effect of a holding time (either in tension or in compression, and over what duration?) without forgetting the effect of the surface state, since the crack initiation phase seems to be the most impacted by the PWR environment. Finally, to complete the proposed methodology (Fen) with longer-term actions, pre-cracking may have an effect, and it would be good to clarify the impact of the dissolved oxygen level as well.

A modification of the thickness of the specimen would also make it possible to obtain a less severe gradient of the bending stresses, which would bring many elements of understanding and quantification to the studies of the propagation of deep cracks with or without environmental effects.

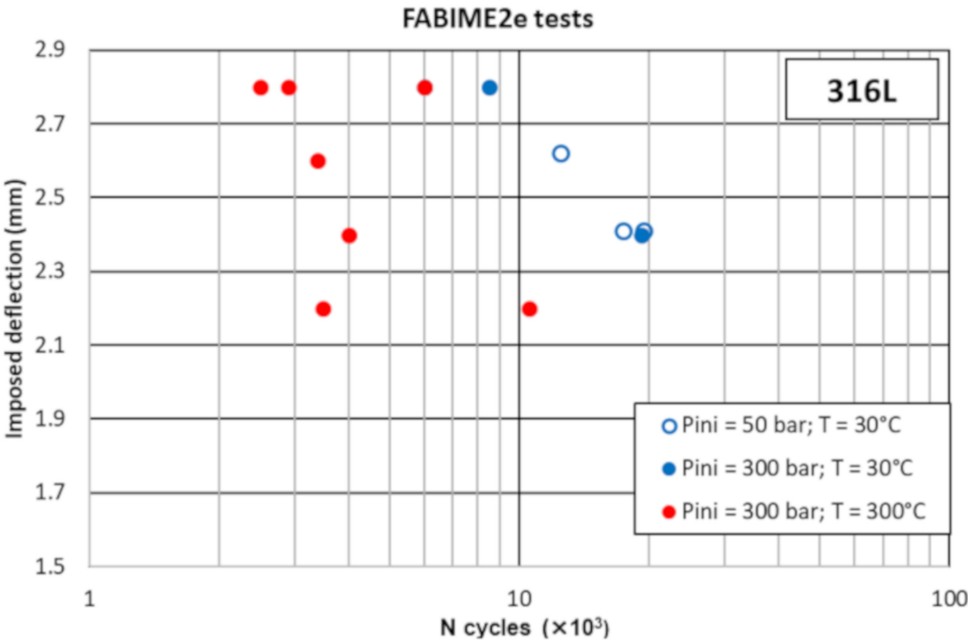

**Figure 14.** Experimental data of spherical fatigue tests obtained with the FABIME2e device for different conditions (temperature, pressure).

### 6. Conclusions

This paper focused on the description of two kinds of experimental devices to perform fatigue tests on spherical specimens with or without the effect of a water environment.

The first device (FABIME2) is devoted to study of the effect of biaxiality and mean strain/stress on fatigue life. Biaxial fatigue tests were carried out on two austenitic stainless steels: 316L THY and 304L CLI. The results obtained show that crack initiation under an equibiaxial load has a low impact on the fatigue life, which remains in the field covered by the design curve defined and used in the codification.

A second and new device named FABIME2e ("e" for environment) was also developed for the study of the impact of the environmental effect. This device studies the impact of the equibiaxial loadings with a primary water environment (PWR, 300 °C with a permanent pressure of 140 bars). The first results obtained with the FABIME2e device showed an aggravating effect of the distilled water environment on fatigue performance.

**Author Contributions:** Conceptualization, C.G.; experimental validation, C.G., H.D. and G.P.; software, G.P.; writing—original draft preparation, C.G.; writing—review, G.P., H.D., L.D.B. and J.-C.L.R. All authors have read and agreed to the published version of the manuscript.

**Funding:** This research received no external funding.

**Institutional Review Board Statement:** Not applicable.

**Informed Consent Statement:** Not applicable.

**Data Availability Statement:** Not applicable.

**Acknowledgments:** F. Datcharry and M. Rousseau from CEA are very greatly acknowledged for their participation in the conception of FABIME2E.

**Conflicts of Interest:** The authors declare no conflict of interest.

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
