# Peer review of "Environmental Effect on Fatigue Crack Initiation under Equi-Biaxial Loading of an Austenitic Stainless Steel"

_metals, doi:10.3390/met11020203_

Round 1

Reviewer 1 Report

The paper presents an interesting solution how to perform fatigue tests in biaxial mode while including also the environmental effect. The experimental results are more or less illustrative, the key advance concerns the experimental devices.

Some notes for improvement:

0) Please check the English first - there are so many authors involved that the errors should not occur so often.

1) Please ensure that the frequent abbreviations, which might seem familiar to you, are understood also by readers, when they are first used - PWR, LISN, LVDT, ANL, RDPE, ...

2) L. 50 : Fen has to be multiplied by the fatigue usage factor? Would not it be more consistent vice versa, i.e. the usage factor to be multiplied by Fen?

3) Figs. 1 and 2 - Air curve is a very strange term

4) L. 71-72: The explanation of R=0 and R=-1 got weird and rewording could improve the comprehensibility

5) I wonder, why the specimen is called spherical, when it is obviously circular. This is some common designation in nuclear industry for such specimen types?

6) Figure 3: I understand what you wanted to draw, but I am not sure the figures are clear enough. At least the deformed membranes should be in some way explained...

7) I skipped most of my comments on English usage, but L.106 "The application of under..." seems to me already too odd. Reference for those directives is missing.

8) Clarify, whether you want to capitalize first letters in heading, or you don't want to do that. Mixing both approaches is not nice.

9) I miss the reference for RSE-M, which is noted here (L. 113, L. 115)

10) Why is TRESCA capitalized? This was his name. He was French, so this is the reason? :-)

11) L. 140: I guess you should refer there to Figure 5

12) L. 141-142: First, the formulation "... aggravating effect of one condition on the other condition" does not make a sense due to wording chosen.

Second, the illustration in Figure 5 deserves a bit more complex description. You are presenting only a plenty of points stating (probably) that equibiaxial load condition is providing worse results when compared with uniaxial conditions. There are no regression lines (probably Stromeyer, Bastenaire or Kohout-Věchet would be necessary for uniaxial tests), but it seems to me that equibiaxial tests reach higher lifetimes for 316L and lower lifetimes for 304L-CLI if compared with their uniaxial versions. So, technically, one of them should aggravate the situation.

Third, the graph shows how effective is the von Mises condition in the conversion of the biaxial state to uniaxial. I am not sure at all, if this output can be called "effect".

13) L. 151 - 340 degC is mentioned, L. 189 - only 240 degC stated. I guess this is a typo.

14) L. 156: I haven't heard about "oil fatigue device" before, maybe other wording would help.

15) L. 173: Complaisance seems to be too French, what about compliance?

16) Figure 8: It is nice for illustration, but I guess the readers would be interested in dimensions of the specimens tested - either here or in the text.

17) Figures 4, 6 and 11: Better resolution of the figures would help.

18) Figure 12: The figures are so tiny, that the only dimension shown there is not legible. If it is not necessary, why it is there; if it is necessary, the figures can be enlarged.

19) Figure 13: For a surface crack of 5 mm size, the applied stress decreased approximately for the uniaxial loading, but not for equibiaxial. I am not sure I understand the consequence - the stress decrease by 20% will be used for FABIME2? I do not see any such explanation.

20) L. 290: What is side 2?

21) L. 301-302: "...life is ... reduced ... between the two temperatures..." I would expect it would reduce when going from one condition to another one, not between.

22) Figure 16: Pini=50-200 bar - it is not clear which pressure was used

23) L. 341: Where we could see this aggravating effect? There is no direct comparison here. Should not it be transferred to the section "Discussion"? Conclusion should summarize the results described previously.

24) L. 343-356: It seems to me that this also belongs to Discussion. Not much is then left for Conclusion, so reflect about it.

Author Response

Response ti the reviewer's comments :

0) Please check the English first - there are so many authors involved that the errors should not occur so often.

1) Please ensure that the frequent abbreviations, which might seem familiar to you, are understood also by readers, when they are first used - PWR, LISN, LVDT, ANL, RDPE,

=>Done in L-18 for PWR, L-24 for LISN, L-25 for EDF, L 69 for ANL, L-100 for CEA, L-103 for LVDT,

 2) L. 50 : Fen has to be multiplied by the fatigue usage factor? Would not it be more consistent vice versa, i.e. the usage factor to be multiplied by Fen?

=> It depends on the point of view, but both are possible. The general definition is  Uen = Fen * U

 3) Figs. 1 and 2 - Air curve is a very strange term . 

=>Figures are extract for literature, the best fit air curve are defined by ANL and uses their own model.

 4) L. 71-72: The explanation of R=0 and R=-1 got weird and rewording could improve the comprehensibility

  • L 80-81, explanation of load ratio extended

 5) I wonder, why the specimen is called spherical, when it is obviously circular. This is some common designation in nuclear industry for such specimen types?

=>The spherical denomination is related to the final geometry when it is strained at the end of the tests. This shape represents a spherical part.

 6) Figure 3: I understand what you wanted to draw, but I am not sure the figures are clear enough. At least the deformed membranes should be in some way explained...

  • L90, Figure change

 7) I skipped most of my comments on English usage, but L.106 "The application of under..." seems to me already too odd. Reference for those directives is missing.

  • L 117, The sentence changes

 8) Clarify, whether you want to capitalize first letters in heading, or you don't want to do that. Mixing both approaches is not nice.

 => Done

9) I miss the reference for RSE-M, which is noted here (L. 113, L. 115)

 => L 402 403, Ref is added

10) Why is TRESCA capitalized? This was his name. He was French, so this is the reason? :-)

 => L 146, No typo error

11) L. 140: I guess you should refer there to Figure 5

 => L 152, Ref added

12) L. 141-142: First, the formulation "... aggravating effect of one condition on the other condition" does not make a sense due to wording chosen.

=> L 153-157, The sentence was modified.

Second, the illustration in Figure 5 deserves a bit more complex description. You are presenting only a plenty of points stating (probably) that equibiaxial load condition is providing worse results when compared with uniaxial conditions. There are no regression lines (probably Stromeyer, Bastenaire or Kohout-Věchet would be necessary for uniaxial tests), but it seems to me that equibiaxial tests reach higher lifetimes for 316L and lower lifetimes for 304L-CLI if compared with their uniaxial versions. So, technically, one of them should aggravate the situation.

Third, the graph shows how effective is the von Mises condition in the conversion of the biaxial state to uniaxial. I am not sure at all, if this output can be called "effect".

  • Comment: We took the point of view of presenting the experimental data directly, without proposing any smoothing curve.

To this, it should be added that the equibiaxial tests were carried out in an oil environment (therefore with very little oxygen) and that the uniaxial tests were carried out in air (with oxygen). However, it has been shown that fatigue tests carried out in an environment without air (under vacuum) had a longer life span.

It is for this reason that we consider (proportionally speaking) that the impact of equibiaxiality remains an element of the second order in front of the "natural" dispersion of fatigue tests on austenitic stainless steels

13) L. 151 - 340 degC is mentioned, L. 189 - only 240 degC stated. I guess this is a typo. Yes, The capacities of the device is 400°C max. The PWR conditions are a temperature of 340°C.

=> L203 (add upt to 400) and L 207 Correct 

14) L. 156: I haven't heard about "oil fatigue device" before, maybe other wording would help. Title of part 2 describe the first fatigue device with the oil environnement.

 => L174 : added same name pf the first device

15) L. 173: Complaisance seems to be too French, what about compliance?

 => L191, Yes

16) Figure 8: It is nice for illustration, but I guess the readers would be interested in dimensions of the specimens tested - either here or in the text.

 => L222 Size of the specimen added in the caption

17) Figures 4, 6 and 11: Better resolution of the figures would help.

 => L93, 188, 288, 296, size of the figure are enlarged

18) Figure 12: The figures are so tiny, that the only dimension shown there is not legible. If it is not necessary, why it is there; if it is necessary, the figures can be enlarged.

  => L296, size of the figure are enlarged

19) Figure 13: For a surface crack of 5 mm size, the applied stress decreased approximately for the uniaxial loading, but not for equibiaxial. I am not sure I understand the consequence - the stress decrease by 20% will be used for FABIME2? I do not see any such explanation.

=> In Figure 13, the evolution of the in-plane normal stress in the case of the spherical specimen is shown as a function of crack size. The objective of this figure and of this numerical work is to find a relation between a crack size generating a stress reduction of 25% in the uniaxial case with a crack generating an equivalent reduction of the normal stress at the plane of the crack in the equibiaxial case.

 20) L. 290: What is side 2?

 => L312 : Side 2 to observed side

21) L. 301-302: "...life is ... reduced ... between the two temperatures..." I would expect it would reduce when going from one condition to another one, not between.

=>L 324 Yes, Test done with 30°C are a longer llife than test done with 300°C. Between is too French, I change the formula.

 22) Figure 16: Pini=50-200 bar - it is not clear which pressure was used,

  • L350 : Initial pressure is P=50 bar for the tests (Remainder of other graphics)

23) L. 341: Where we could see this aggravating effect? There is no direct comparison here. Should not it be transferred to the section "Discussion"? Conclusion should summarize the results described previously.

 =>L353-366 : Modification of the conclusion, text change from conclusion to Experimental results

24) L. 343-356: It seems to me that this also belongs to Discussion. Not much is then left for Conclusion, so reflect about it.

 =>L353-366 : Modification of the conclusion, text change from conclusion to Experimental results

Reviewer 2 Report

The paper is well written and explains 2 experimental devices which can be used to perform fatigue tests on spherical specimens.

The paper has novelty and is highly recommended.

Author Response

Response to the reviewer's comments and remarks.

I provide some checks for english language in the revised version of paper.

Reviewer 3 Report

Comments for metals-1068630:

This work presented two kinds of experimental devices for fatigue tests of spherical specimens. Specifically, FABIME2 was developed to account for the biaxiality effect and mean strain/stress effect and FABIME2e (“e” for environment) was developed to investigate the impact of the environmental effect. Based on the two devices, the influence of environment effect on fatigue crack initiation of austenitic stainless steels were investigated. The contents of current work should be of interest for the field, however, the structure of this paper is not so well organized. In addition, some details about the establishment of the proposed model need further clarification and some concepts used in this study should be explained.

  • In Abstract part, the abbreviated words “PWR”, “LISN”, “EDF” represent what? In addition, explain the abbreviated word for the first time it is used in the text.
  • In Abstract part, the author is suggested to present less background and more research motivations.
  • In Introduction Section, research status should be summarized and corresponding research significance of this work should be presented.
  • The qualities of all Figures need to be refined as well as font size.
  • The usage of “equi-biaxial” and “equibiaxial” should be unified. Check full text for similar mistakes.
  • A flow chart should be added to illustrate the steps of fatigue tests.
  • Figure 5 should be introduced in the text and more comments on the results of fatigue tests should be added.
  • In Fig. 7, more details on the FABIME2e should be given, like the units inside;
  • It is suggested that Section 4 and Section 5 should be combined.
  • The conclusions need to be shortened and refined.
  • English usage of full text should be refined as well as paper structure.

Author Response

Response to the reviewer's comments:

In Abstract part, the abbreviated words “PWR”, “LISN”, “EDF” represent what? In addition, explain the abbreviated word for the first time it is used in the text. =>Done in L-18 for PWR, L-24 for LISN, L-25 for EDF, L 69 for ANL, L-100 for CEA, L-103 for LVDT,

In Abstract part, the author is suggested to present less background and more research motivations.

=> L22-23, Motivations are added

In Introduction Section, research status should be summarized and corresponding research significance of this work should be presented.

=> L57-61, Additional and background studies are added to the introduction with

=> L 420-425 with a corresponding references

  • The qualities of all Figures need to be refined as well as font size.

=> L90 188, 288, 296, 303 Figures are upgrading, size are enlarged for some one

  • The usage of “equi-biaxial” and “equibiaxial” should be unified. Check full text for similar mistakes.

=>  Done

  • A flow chart should be added to illustrate the steps of fatigue tests.
  • Figure 5 should be introduced in the text and more comments on the results of fatigue tests should be added.

=>L153-157 Textx are upgrading

  • In Fig. 7, more details on the FABIME2e should be given, like the units inside;

=> L218, Figure change

  • It is suggested that Section 4 and Section 5 should be combined. Done

=>L163, 164 205 : Section 4 and 5 become only section 4

  • The conclusions need to be shortened and refined.

 =>L353-366 : Modification of the conclusion, text change from conclusion to Experimental results

  • English usage of full text should be refined as well as paper structure.

Round 2

Reviewer 1 Report

Most of my objections were solved. I would only note, that you worked in haste, and some new text parts are not separated from the older parts by any space character. Also the part transferred from the Conclusion to the Experimental results was not deleted from Conclusion, so please revise it - otherwise it is left there twice.

Good luck in your next activities.

Author Response

Thank you very much for your review work. I appreciated your feedback.

L381-393: The part of the conclusion that is now in the experimental part has indeed been removed from the conclusion.

Kind regards,

Reviewer 3 Report

All the comments have been addressed in this revision, it can be accepted as it is.

Author Response

Thank you very much for your review work. I appreciated your feedback.
Kind regards,

This manuscript is a resubmission of an earlier submission. The following is a list of the peer review reports and author responses from that submission.